# Morpholine Radical in the Electrochemical Reaction with Quinoline *N*-Oxide

Egor L. Dolengovski [1], Tatyana V. Gryaznova [1], Oleg G. Sinyashin [1], Elena L. Gavrilova [2], Kirill V. Kholin [2] and Yulia H. Budnikova [1,*]

[1] Arbuzov Institute of Organic and Physical Chemistry, FRC Kazan Scientific Center, Russian Academy of Sciences, 8 Arbuzov Street, 420088 Kazan, Russia; dolengovski@gmail.com (E.L.D.); tatyanag@iopc.ru (T.V.G.); oleg@iopc.ru (O.G.S.)

[2] Organic Chemistry Department, Kazan National Research Technological University, 68 Karl Marx Street, 420015 Kazan, Russia; gavrilova_elena_@mail.ru (E.L.G.); kholin06@mail.ru (K.V.K.)

[*] Correspondence: yulia@iopc.ru

**Abstract:** An electrochemical reaction between quinoline *N*-oxides and morpholine was developed by using Cu(OAc)$_2$ as a catalyst, generating products of 4-aminoquinoline *N*-oxides in CH$_2$Cl$_2$ or 2-aminoquinoline *N*-oxides in CH$_3$CN in good yields. With an increase in the amount of electricity passed, the product deoxygenates with the formation of aminoquinolines. The advantages of the reaction are mild conditions, room temperature, the use of morpholine rather than its derivatives, and the ability to control the process when the electrolysis conditions change. Bisubstituted quinoline has also been obtained. The redox properties of both individual participants of C–H/N–H cross-coupling and multicomponent systems were established by voltammetry and EPR methods. For the first time, the EPR spectrum of the morpholine radical was recorded at room temperature, and its magnetic resonance parameters were determined in CH$_2$Cl$_2$. Mechanisms for the catalytic reaction have been proposed. This is a simple and easy-to-perform method for introducing a morpholine substituent, important in medicinal chemistry and other fields, by C–H/N–H cross-coupling.

**Keywords:** copper; catalysis; radical; morpholine; *N*-oxides; electrochemistry; EPR; C–H functionalization; cross-coupling

## 1. Introduction

Morpholine is considered a favored scaffold for medicinal chemistry [1–6]. Synthetic and natural molecules bearing a morpholine substituent have multiple biological activities, as well as improved pharmacokinetic and metabolic profiles, which have placed morpholine as one of the most promising compounds evaluated in structure–activity relationship studies (Figure 1). The advantage of morpholine over other nitrogen-containing heterocycles is its electron-deficient ring due to the negative influence of oxygen, as well as the relatively lower basicity of nitrogen.

Due to its biological and pharmacological significance, the synthesis of morpholines has been extensively studied in recent years and requires new approaches, especially synthetic routes towards the late-stage functionalization or introduction of this moiety. Little is known about simple one-step methods for C–H/N–H cross-coupling of morpholine with organic molecules.

Attempts to involve morpholine in radical addition reactions at multiple bonds were described as early as 1963 [7]. It was assumed that heating at elevated temperatures in the presence of t-butyl peroxide would make it possible to generate radicals of morpholine and other cyclic esters (dioxane), but the efficiency of this approach turned out to be low, and the yields of products, both with preservation of the morpholine cycle and acyclicity, are low (Scheme 1a).

**Figure 1.** Representative examples of morpholine-based drugs from [2,3].

The existence and key role of the morpholine radical have been suggested in a number of studies, for example, photoredox/nickel catalysis of aryl halides and alkyl boronic ester cross-coupling [8] (Scheme 1b). In the absence of morpholine (or other alkyl amines, although morpholine proved to be the best amino radical transfer (ART) reagent), the reaction does not proceed at all. It is assumed that [Ir(dF(CF$_3$)-ppy)$_2$(dtbbpy)]PF$_6$ as photocatalyst reacts with morpholine to form its radical. The latter acts as an amine radical transfer reagent, activates boronic esters, generates alkyl radicals from them, and then the catalytic cycle already includes the transformation of nickel complexes and their sigma-alkylnickel intermediates. The oxidation of morpholine in DMF comes easily, $E_{1/2}(D^{\bullet+}/D) = +1.18$ V vs. SCE [8], and is the first step of the photocatalytic reductive quenching cycle, furnishing the N-centered radical and the reduced Ir$^{II}$ complex. The C(sp$^2$)−C(sp$^3$) coupling reaction was fully inhibited in the presence of TEMPO, confirming the existence of an alkyl radical intermediate derived from a boronic reagent, which was isolated with a 31% yield of TEMPO hydroxylamine adduct. Interestingly, no adducts with morpholine or its radical have been recorded, and there is no EPR evidence in this work.

Morpholine-based RAFT (reversible addition–fragmentation chain-transfer) agents are used for the reversible deactivation radical polymerization [9], possibly due to morpholine radical intermediates that are not fixed.

There are few descriptions of electrochemical syntheses using morpholine or its derivatives. The key reaction in the synthesis of isomeric 3-oxadiazolyl/triazolyl morpholines was an electrochemical C–H oxidation/metoxylation of N-Boc morpholine [10] (Scheme 1c).

Cobalt [11] or nickel [12] catalyzed C–H amination of arenes by morpholine has been developed in electrochemical oxidative conditions (Scheme 1d).

Although the authors of [11] made assumptions about the reaction mechanism based on the oxidation of Co$^{II}$ to Co$^{III}$, it was not confirmed and was not specially studied, and the redox properties of morpholine and the stage with its participation were not established. In [12], the oxidation potential of morpholine was determined as ~0.7 V ref. Fc/Fc$^+$ in DMA, and the mechanism includes the Ni(II)/Ni(III)/Ni(IV) catalytic cycle due to the formation of a nickel cycle with a benzamide substrate, which is oxidized earlier morpholine. Thus, it was assumed that morpholine is not oxidized at the electrode in these nickel- or cobalt-catalyzed transformations. O-benzoylhydroxylamines in the presence of CuCl as a catalyst in 1,2-dichloroethane (DCE) at 80 °C afford 2-aminoquinolines [13], and Cu(I) has been hypothesized to induce splitting of an amine precursor to the morpholine radical (Scheme 1e), but this has not been confirmed.

Methods for the synthesis of quinoline derivatives have been of particular interest in recent years [14–16]. Quinoline *N*-oxides as partners of C–H/N–H coupling and C–N bond formation in this regard have not been sufficiently studied, including electrochemically.

In this work, we propose a new method for introducing a morpholine substituent into quinoline *N*-oxide under mild electrocatalytic conditions with the participation of Cu(OAc)$_2$ as a catalyst at room temperature. The oxidation of morpholine at the electrode, its redox properties in the absence and presence of C–H/N–H coupling partners (morpholine and *N*-oxide), and the EPR characteristics of the morpholine radical are also described.

**Scheme 1.** Selected known reactions of morpholine, including those involving its putative radical: (**a**) Walsh, 1963 [7]; (**b**) Maier, 2022 [8]; (**c**) Mykhailiuk, 2017 [10]; (**d**) Lei, 2018 [11], Ackermann, 2018 [12]; (**e**) Wang, 2018 [13].

## 2. Results

### 2.1. Electrochemical Synthesis

The electrochemical oxidative amination of quinoline *N*-oxide (**1**) using morpholine (**2**) as an aminating reagent in the presence of a copper or silver salt catalyst (10 mol.%) has been studied (Scheme 2). The electrolysis was carried out in a three-electrode cell with separation of the anode and cathode spaces at room temperature without a background electrolyte in the anode space (although the addition of $K_3PO_4$ base in some cases (lines 4–7) lowered the potential and improved the conductivity) in a galvanostatic mode. However, the electrolysis potential was controlled to establish a mechanism (the reference electrode is Ag/AgNO$_3$). The cathode space contained a saturated solution of PyHBF$_4$ in an appropriate solvent. Cu(OAc)$_2$ and AgOAc were tested as catalysts. All copper-catalyzed syntheses proceeded with 100% conversion of the initial reagents. The results of electrosynthesis are summarized in Table 1. Analysis of the NMR and mass spectra of the products isolated

by column chromatography showed that morpholine adds to quinoline *N*-oxide, giving different products depending on the synthesis conditions. When potassium triphosphate is used as a base, the yield of ortho-coupling products reaches 80% (Table 1, line 4). Electrolysis in the undivided cell was not successful. The best catalyst is Cu(OAc)$_2$. AgOAc is not effective enough. Interestingly, the replacement of CH$_3$CN with CH$_2$Cl$_2$ changed the direction of morpholine functionalization to the para position while retaining the *N*-oxide moiety (**3**) (Table 1, line 7). In acetonitrile, an increase in the electrolysis time, i.e., the number of electrons passed, in all cases led to a deoxygenation product, the formation of substituted quinoline (products **3**, **4**, and **6**, lines 2, 3, and 5–7, respectively; Cu(OAc)$_2$ catalyst). In these deoxygenation reactions, morpholine *N*-oxide (**7**) was isolated as a by-product, taking oxygen from quinoline *N*-oxide.

**Scheme 2.** Electrochemical coupling of morpholine with quinoline *N*-oxide.

**Table 1.** Electrochemical cross-coupling products of morpholine and quinoline *N*-oxide, 60 mA.

| No. | Ratio Morpholine: *N*-Oxide | Catalyst | Solvent | Base | Number of Electrons (Faradays) | Potential, V | Product (Yield, %) |
|---|---|---|---|---|---|---|---|
| 1 | 1.2:1 | Cu(OAc)$_2$ | CH$_3$CN | - | 2 | 1.80 | **5** (54) |
| 2 | 1.2:1 | Cu(OAc)$_2$ | CH$_3$CN | - | 3 | 2.00 | **6** (31) |
| 3 | 2.4:1 | Cu(OAc)$_2$ | CH$_3$CN | - | 4 | 2.20 | **4** (48) |
| 4 | 1.2:1 | Cu(OAc)$_2$ | CH$_3$CN | K$_3$PO$_4$ | 2 | 1.49 | **5** (80) |
| 5 | 1.2:1 | Cu(OAc)$_2$ | CH$_3$CN | K$_3$PO$_4$ | 3 | 1.80 | **6** (67) |
| 6 | 3:1 | Cu(OAc)$_2$ | CH$_3$CN | K$_3$PO$_4$ | 4 | 1.90 | **4** (52) |
| 7 * | 3:1 | Cu(OAc)$_2$ | CH$_2$Cl$_2$ | K$_3$PO$_4$ | 4 | 0.30 | **3** (64) |
| 8 | 1.2:1 | AgOAc | CH$_3$CN | - | 2 | 1.84 | **5** (32) |
| 9 | 1.2:1 | AgOAc | CH$_3$CN | - | 3 | 2.08 | **6** (24) |
| 10 | 3:1 | AgOAc | CH$_3$CN | - | 4 | 2.10 | **4** (46) |

* Additive of Et$_4$NBF$_4$ (0.02 M) was used for electroconductivity.

We tested other aromatic *N*-oxides (pyridine *N*-oxide, phenylpyridine *N*-oxide, *iso*-quinoline *N*-oxide), and the results are listed in the Supplementary Materials. Of the studied *N*-oxides, *iso*-quinoline *N*-oxide showed high yields in the morpholination reaction, while others were either non-reactive or the reaction proceeded non-selectively (Supplementary Materials Table S1).

Deoxygenation reactions of *N*-oxides, including quinoline *N*-oxide, usually take place at reductive electrochemical conditions [17,18] and during the substitution of C–H bonds, for example, in [19] where Ph$_3$P additives and PhI(OAc)$_2$ oxidant are necessary for the amidation of quinoline *N*-oxides with arylsulfonamides and favor this reaction at 80 °C. Deoxygenative C-2 amination of quinoline *N*-oxides is observed in the reaction between quinoline *N*-oxides and O-benzoylhydroxylamines using CuCl as a catalyst [13]. 1,2-Dichloroethane (DCE) as a solvent served as a reducing agent to cleave the N–O bond [13]. Obviously, under our oxidizing conditions, none of the mechanisms proposed in these works can be implemented since we do not use Ph$_3$P and the oxidizing agent PhI(OAc)$_2$, nor CuCl or dichloroethane as a reductant.

### 2.2. Voltammetry Data

The oxidation potentials of the participants in the studied transformations are shown in Tables 2 and 3 and the corresponding Figure 2 (CH$_3$CN) and Figure 3 (CH$_2$Cl$_2$).

**Table 2.** Oxidation potentials of key compounds in CH$_3$CN.

| Entry | 1st Peak (vs. Fc/Fc$^+$) | 2nd Peak (vs. Fc/Fc$^+$) | 3rd Peak (vs. Fc/Fc$^+$) |
|---|---|---|---|
| Quinoline N-ox | 1.10 V | - | - |
| Cu(OAc)$_2$ | 1.41 V | 1.87 V | - |
| Morpholine | 0.85 V | - | - |
| Quinoline N-ox + Cu(OAc)$_2$ | 1.24 V | - | - |
| Morpholine + Cu(OAc)$_2$ | 0.74 V | 1.32 V | 1.83 V |
| Quinoline N-ox + Morpholine + Cu(OAc)$_2$ | 0.74 V | 1.29 V | - |

**Table 3.** Oxidation potentials of key compounds in CH$_2$Cl$_2$.

| Entry | 1st Peak (vs. Fc/Fc$^+$) | 2nd Peak (vs. Fc/Fc$^+$) | 3rd Peak (vs. Fc/Fc$^+$) |
|---|---|---|---|
| Quinoline N-ox | 1.18 V | - | - |
| Cu(OAc)$_2$ | - | - | - |
| Morpholine | 0.89 V | - | - |
| Quinoline N-ox + Cu(OAc)$_2$ | 1.16 V | 1.81 V | - |
| Morpholine + Cu(OAc)$_2$ | 0.85 V | 1.55 V | - |
| Quinoline N-ox + Morpholine + Cu(OAc)$_2$ | 0.84 V | 1.29 V | 1.63 V |

Morpholine oxidizes more easily than quinoline *N*-oxide or Cu(OAc)$_2$ in CH$_3$CN (Figure 2). Mixtures of reaction participants do not show any catalytic currents in CH$_3$CN. The little shifts in the oxidation potentials of the participants due to possible complex formation between them are observed in some cases (E$_p$ of morpholine, copper, and quinoline *N*-oxide oxidations are shifted to less positive potentials by ~0.1 V after mixing). The color of the copper salt solution changes from light blue to green when morpholine is added.

In CH$_2$Cl$_2$, the first oxidation peak of the morpholine–quinoline *N*-oxide–Cu(OAc)$_2$ reaction mixture also corresponds to the oxidation peak of pure morpholine, but the subsequent peaks and their currents are noticeably different. Cu(OAc)$_2$ in CH$_2$Cl$_2$ does not give a clear oxidation peak; in contrast to CH$_3$CN, it is characterized by a diffuse wave at higher potentials, possibly due to very low solubility. However, in the presence of *N*-oxide or morpholine partners, solubility improves significantly.

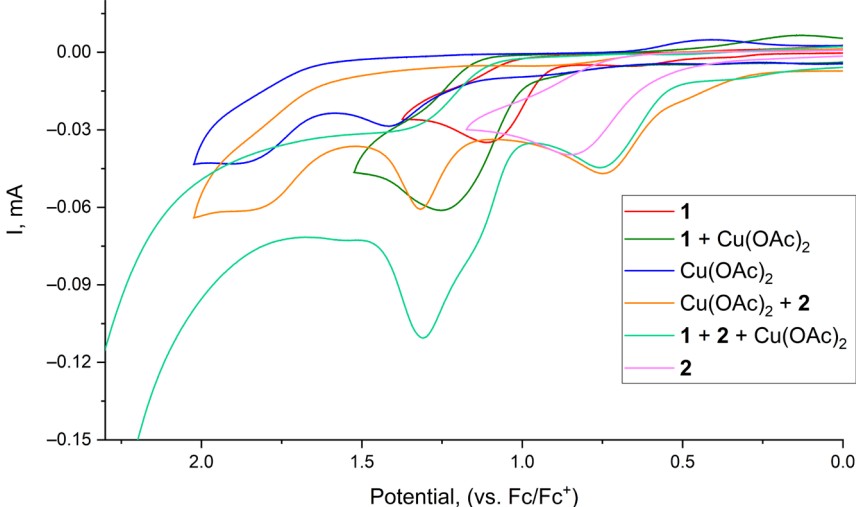

**Figure 2.** CV of quinoline *N*-oxide (**1**), morpholine (**2**), Cu(OAc)$_2$, and their mixtures in CH$_3$CN. Bu$_4$NBF$_4$ 0.1 M; CH$_3$CN; working/auxiliary electrode—Pt/Pt; all compounds $5 \times 10^{-3}$ M concentration; 100 mV/s; Ag/AgNO$_3$ as reference electrode (recalculated against Fc/Fc$^+$).

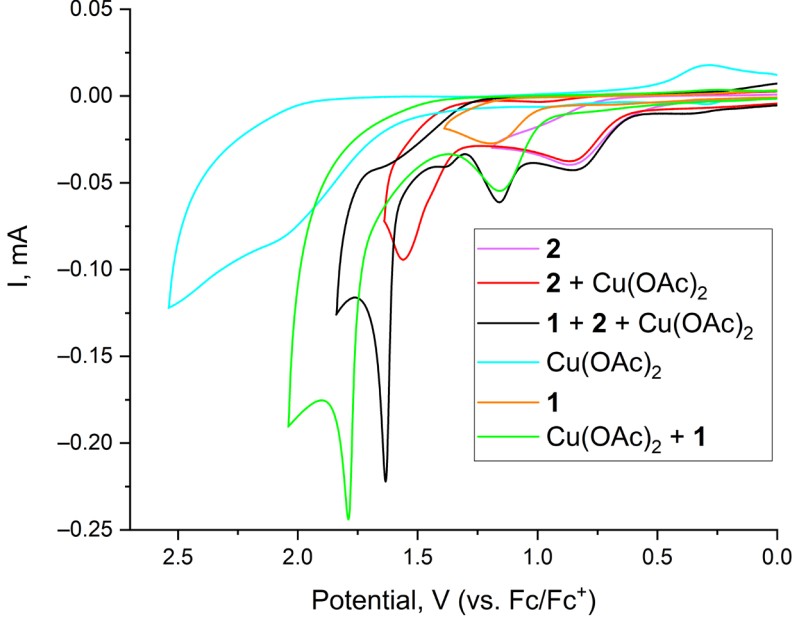

**Figure 3.** CV of quinoline *N*-oxide (**1**), morpholine (**2**), Cu(OAc)$_2$, and their mixtures in CH$_2$Cl$_2$. Bu$_4$NBF$_4$ 0.1 M; CH$_2$Cl$_2$; working/auxiliary electrode—Pt/Pt; all compounds $5 \times 10^{-3}$ M concentration; 100 mV/s; Ag/AgNO$_3$ as reference electrode (recalculated against Fc/Fc$^+$).

In acetonitrile, the second peak of the ternary mixture has a complex character due to the merging and superposition of the oxidation peaks of quinoline oxide and copper, and its current approximately corresponds to two electrons. The preparative electrolysis proceeds just at the potential of this last peak (Table 2).

In CH$_2$Cl$_2$, the second oxidation peak of the ternary mixture corresponds to that of pure quinoline oxide; however, a new peak appears with a higher oxidation current at higher potentials, 1.63 V (Figure 3).

Therefore, for the electrolysis potential required to implement the process of cross-coupling of morpholine and quinoline *N*-oxide, competitive oxidation of each participant is apparently possible (Figure 2, CH$_3$CN). However, in CH$_2$Cl$_2$, the electrolysis potential is much lower, so it should be assumed that morpholine is preferably oxidized in this case (Table 1, line 7).

### 2.3. EPR Study

During the oxidation of a solution of morpholine in methylene chloride at room temperature at a potential of +0.6 V (in the e-EPR cell), the spectrum shown in Figure 4 appears and grows. As the electrode potential increases, the EPR spectrum changes and disappears.

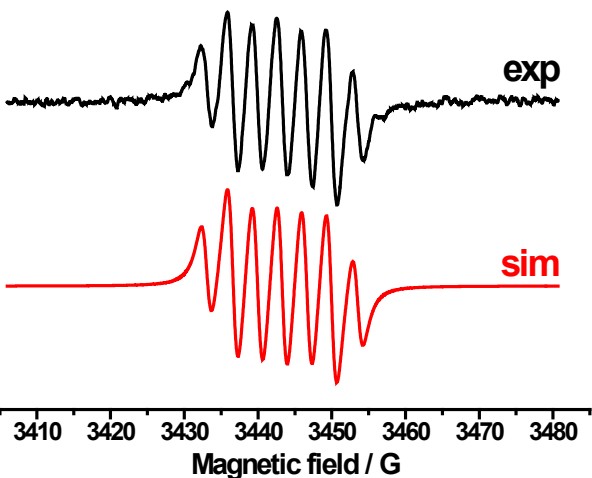

**Figure 4.** EPR spectrum registered during morpholine (**2**) oxidation with $5 \times 10^{-3}$ M concentration; $CH_2Cl_2$, 0.1 M $Bu_4NBF_4$, rt.

The simulation of the obtained spectrum showed the highest correlation at the following magnetic resonance parameters: g = 2.0069, aN = 6.70 G, 2:aH = 3.55 G.

In the presence of $Cu(OAc)_2$, the signals of radicals from morpholine are not detected in the EPR spectrum, but the Cu(II) signal is initially present; at a potential of about 1.0 V, this signal disappears, apparently due to the oxidation of copper to Cu(III).

According to our data, there are no EPR spectra and corresponding characteristics of the morpholine radical recorded in the literature under normal conditions and in the absence of non-indifferent oxidizing agents. EPR spectrum of frozen solution of morpholine (0.3%) in freon-11 irradiated with X-rays (Philips X-ray generator) or 60Co γ-rays (K-120000 apparatus) was recorded at 155 K, and simulated spectrum of morpholine radical cation was proposed with the proton hyperfine splittings [20]. G-tensor has not been defined.

Earlier, in 1968 [21], the EPR spectrum of morpholine nitroxide, obtained from morpholine in the presence of the hydrogen peroxide radical initiator (100 vol.), which converts morpholine to morpholine *N*-oxide, was recorded in aqueous solution. Hyperfine coupling constants aN = 18.1 G, 2:aH = 12.5 G are several times higher than those observed in this study and rightly refer not to morpholine radical or radical cation but to its *N*-oxide.

Initially, the $Cu(OAc)_2$ solution does not deliver EPR spectra, which indicates that the Cu(II) is in a dimeric form in the solution. Earlier, it was shown that the EPR signal of the copper (II) acetate dimer $[Cu_2(OAc)_4(H_2O)_2]$ in $CH_3CN$ was undetectable at room temperature (weak ERP) [22].

An EPR study of morpholine with $Cu(OAc)_2$ mixture in acetonitrile or $CH_2Cl_2$ showed that the solution gives an EPR spectrum even without applying a potential to the working electrode (Figures 5 and 6). The spectrum consists of four equidistant lines due to splitting at the copper nucleus with a nuclear spin of 3/2. The line width decreases with increasing values of the magnetic induction of the external field (decrease in the g factor). Similar spectra are characteristic of low-spin Cu(II) $3d^9$ complexes. The magnetic resonance spectra obtained as a result of spectrum simulation are presented below.

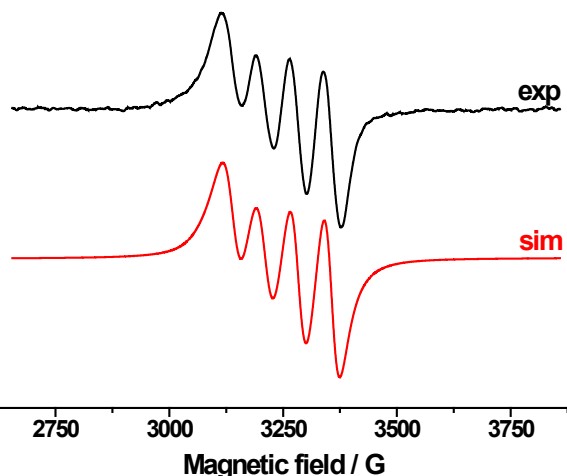

**Figure 5.** EPR spectrum registered during morpholine (**2**) and Cu(OAc)$_2$ oxidation with $5 \times 10^{-3}$ M concentration of both compounds; CH$_3$CN, 0.1 M Bu$_4$NBF$_4$, rt. g = 2.129, a$_{Cu}$ = 72 G, <$\Delta$H> = 39 G (CH$_3$CN).

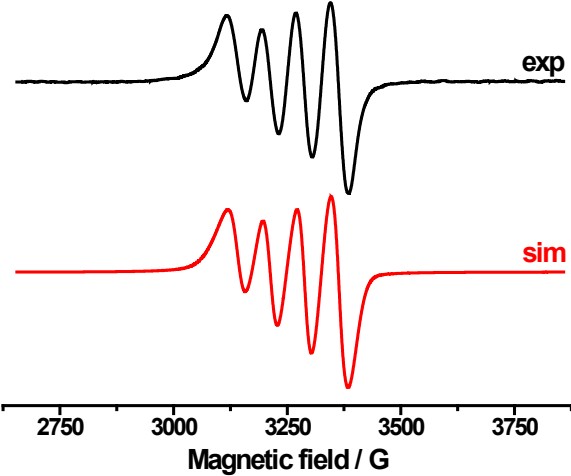

**Figure 6.** EPR spectrum registered during morpholine (**2**) and Cu(OAc)$_2$ oxidation with $5 \times 10^{-3}$ M concentration of both compounds; CH$_2$Cl$_2$, 0.1 M Bu$_4$NBF$_4$, rt. g = 2.127, a$_{Cu}$ = 74 G, <$\Delta$H> = 30 G (CH$_2$Cl$_2$).

The EPR spectrum observed for Cu(OAc)$_2$ is close to previously published [23–25]. The very fact of observing a Cu(II) EPR signal in a morpholine solution indicates the coordination of morpholine with copper, which destroys copper acetate dimers.

When a potential is applied to the electrode, no signals of morpholine radicals are observed, but the intensity of the Cu(II) signal decreases, and the higher the potential, the stronger. This signal disappears at a potential of +1.2 V in CH$_3$CN and decreases at +1.8 V in CH$_2$Cl$_2$ (not completely). Morpholine + quinoline *N*-oxide mixtures did not give EPR signals during oxidation in both solvents (CH$_3$CN, CH$_2$Cl$_2$).

The morpholine radical was not detected in CH$_3$CN. The reasons for the absence of a signal are difficult to determine unambiguously, even in a solution of pure morpholine. It is known that the EPR parameters in various solvents differ even for stable paramagnetic particles, especially the reactivity and lifetime of active radicals should differ. Probably, CH$_2$Cl$_2$ increases the stability of the morpholine radical or changes the direction of oxidation of intermediates at electrolysis potentials.

### 2.4. Mechanistic Considerations

Based on the data from cyclic voltammetry, electron paramagnetic resonance, and preparative synthesis, we proposed schemes for the amination of quinoline *N*-oxide (Schemes 3 and 4).

The reaction is triggered by the electrochemical oxidation of morpholine at the anode with the formation of the morpholine radical, which is detected by the EPR method in $CH_2Cl_2$. The corresponding spectrum is shown above in Figure 2. In the presence of a copper (II) salt, the interaction of morpholine radicals with it is observed since the latter disappears, as does the signal of copper (II) itself gradually disappear in the EPR spectrum. One can assume the formation of a Cu–N complex of morpholine with copper (Scheme 3), which acts as a reagent replacing hydrogen in quinoline *N*-oxide and yields amination to the *para* position of the quinoline ring product in $CH_2Cl_2$.

Since morpholine radicals are not observed in acetonitrile, the mechanism must be different. The stage of copper acetate monomerization in the presence of morpholine is followed by the oxidation of a mixture of cross-coupling precursors and a copper catalyst, as can be assumed, with the formation of $Cu^{III}$, and then the reductive elimination stage leads to the formation of a C–N product of ortho-substitution in *N*-oxide, and $Cu^{I}$ is further oxidized at the anode, and the cycle closes.

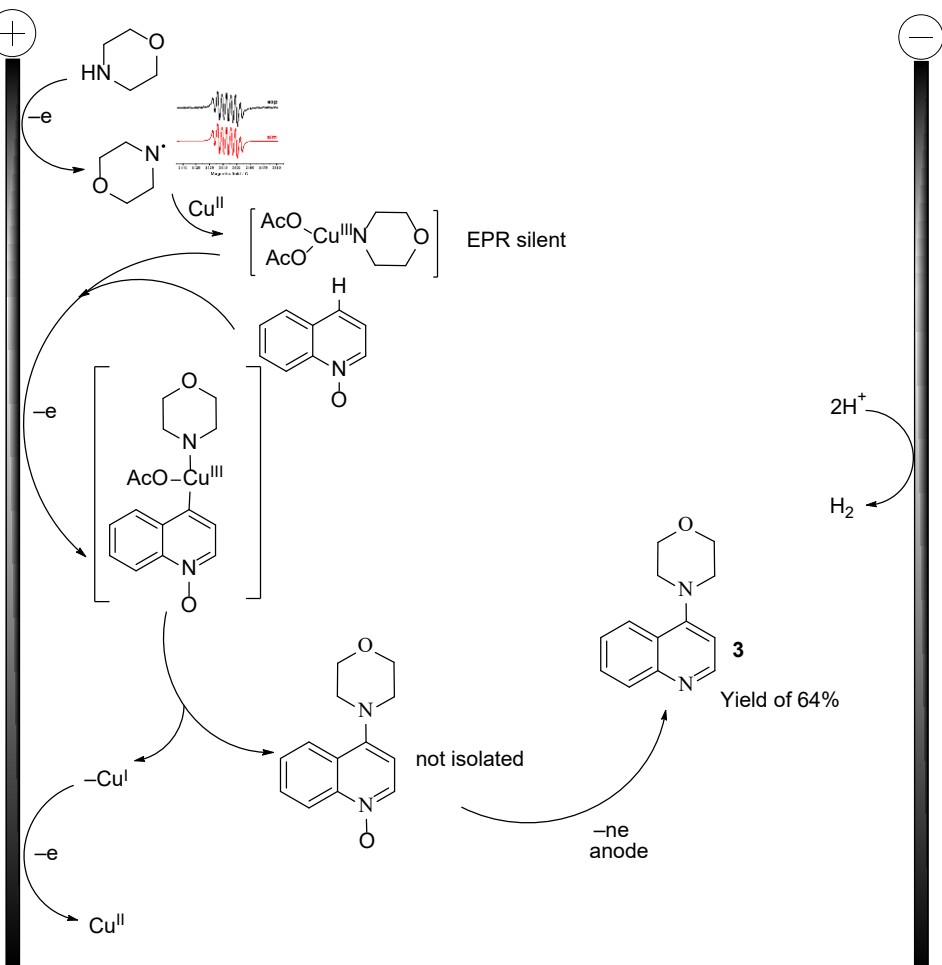

**Scheme 3.** Proposed mechanism of C–N bond formation in $CH_2Cl_2$.

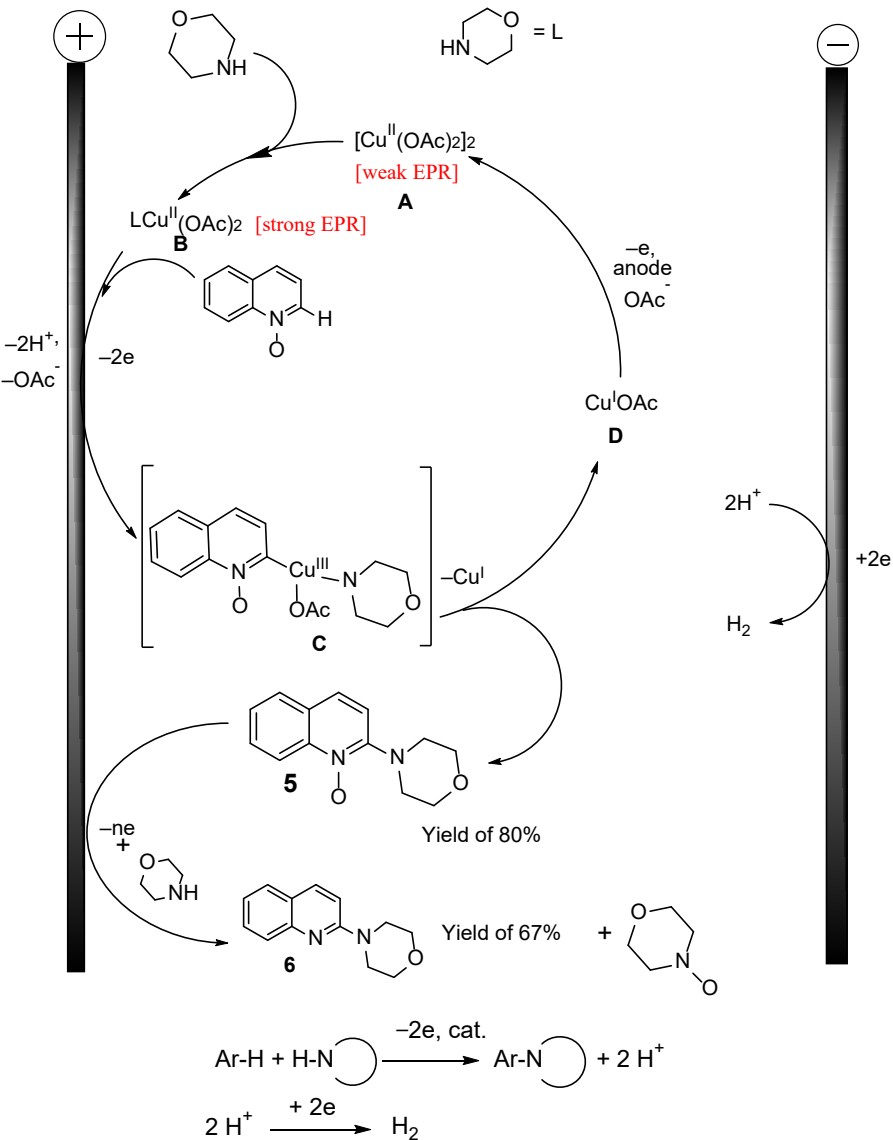

**Scheme 4.** Proposed mechanism of C–N bond formation in CH$_3$CN.

The formation of a copper–carbon bond, or metallization of quinoline *N*-oxide (or another heterocycle) with Cu(OAc)$_2$ during the activation of the C–H bond via copper catalysis, was assumed in many cases [26–28]. However, no one succeeded in isolating and characterizing it in its pure form, as well as us, although we tried to obtain such sigma organometallic compounds with copper in different oxidation states.

An excess of morpholine and an increase in the amount of passed electricity leads to the disubstitution of quinoline *N*-oxide and deoxygenation of the product (**3**, **4**, and **6** in Scheme 1). Morpholin-4-ol (**7**) was found as a by-product, apparently scavenging oxygen from quinoline *N*-oxide (Scheme 1). Generally, the oxidation state of redox-active metals can change during the catalytic cycle; in our case, Cu(I)/(II)/(III) transitions are possible. However, the formation of radicals at individual stages is also possible and proven, and these radicals can not only add themselves to the cross-coupling partner to form a C–N bond but also react with catalyst metals to form metal–nitrogen bonds, changing the oxidation state of the metal, etc. The mechanism of metal–radical catalytic processes is complex and requires further research.

## 3. Materials and Methods

### 3.1. Materials

Quinoline *N*-oxide of 98% purity, $Cu(OAc)_2$ of 99% purity, and anhydrous potassium phosphate tribasic of 97% purity were procured from Acros Organics (Geel, Belgium) without prior purification steps. The supporting electrolytes, $Bu_4NBF_4$ and $Et_4NBF_4$, were obtained from Sigma-Aldrich (St. Louis, MO, USA), 99% purity, and used as received for electrochemical analysis and synthesis. $PyHBF_4$ was used as electrolyte in cathodic part of electrolyzer; the synthesis of $PyHBF_4$ was based on the reaction of pyridine with $HBF_4$ aqueous solution in ethyl alcohol. Morpholine was purified through distillation over KOH. Acetonitrile (CHROMASOLV® Plus) of ≥99.9% purity from Acros Organics was chosen as the primary solvent for the synthesis procedures and voltammetry. The solvent underwent triple distillation, first using potassium permanganate and then over phosphorus pentoxide. Similarly, $CH_2Cl_2$ (ACS reagent, ≥99.9% from Sigma-Aldrich) was purified via distillation over $P_2O_5$.

### 3.2. NMR Measurements

NMR investigations were performed employing Bruker AVANCE-400 (399.93 MHz ($^1$H), 100.6 MHz ($^{13}$C) and Bruker AVANCE-600 (600.1 MHz ($^1$H), 150.9 MHz ($^{13}$C) instruments (from Karlsruhe, Germany). These spectrometers were enhanced with a pulsed gradient unit that has the capability to generate magnetic field pulse gradients in the z-direction, reaching 53.5 G cm$^{-1}$. The chemical shifts are given on the δ (ppm) scale, referenced to the residual solvent signals in both $^1$H and $^{13}$C NMR spectra.

### 3.3. Mass Spectrometric Studies

The products underwent ESI mass spectrometry analysis using an AmazonX spectrometer manufactured by Bruker Daltonik GmbH (from Karlsruhe, Germany). The mass spectra were acquired in the positive ionization mode, employing a capillary voltage set at 4500 V.

### 3.4. EPR Experiments

To eliminate oxygen from liquid samples, a meticulous process involving three cycles of "freezing in liquid nitrogen-evacuation-thawing" was executed. Following the final cycle, the electrolysis cell was purged with gaseous helium. The auxiliary electrode was constructed from platinum, while the reference electrode utilized an $Ag/AgNO_3$ configuration equipped with a bridge made of carbon slate pencil. A working electrode with a diameter of 0.5 mm was crafted from gold wire. The measurements were conducted using a sophisticated apparatus program complex, which included an analog electrochemical system featuring a potentiostat and a PWR-3 programmer, an ELEXSYS E500 ESR spectrometer (Bruker from Karlsruhe, Germany) operating in the X-range, and an E14-440 analog-to-digital and digital-to-analog module (L-Card). These instruments were integrated into a fourth-generation computer system, all working in concert with a distinctive three-electrode helical cell. The ESR spectra were meticulously simulated utilizing the WinSim 0.96 program, a software tool developed by NIEHS for this purpose.

### 3.5. Cyclic Voltammetry (CV)

BASi Epsilon potentiostat sourced from the USA (BASi, West Lafayette, IN, USA) was used for the cyclic voltammogram registration at room temperature conditions. The solution medium comprised acetonitrile or DCM with substrate concentrations set at either 2.5 or $5 \times 10^{-3}$ M. To facilitate the electrochemical processes, $Bu_4NBF_4$ (0.1 M) was employed as the supporting electrolyte. The working electrode was platinum wire of 1.6 mm diameter. Complementing this setup, a platinum rod functioned as the auxiliary electrode. For consistency, all potential measurements were made relative to the $Ag/AgNO_3$ reference electrode and recalculated with respect to $Fe/Fe^+$. Cyclic voltammograms were taken at a scan rate of 100 mV/s.

### 3.6. Preparative Electrolysis

Preparative electrolysis was carried out in galvanostatic mode, but simultaneous control of the working electrode potential was conducted. The 30 mL cylindrical divided cell was used as electrolizer under precise thermostatic conditions. B5-49 direct current source was used at a current of 60 mA ($3 \, mA/cm^2$). The membrane was a ceramic plate with a pore size of 900 nm. The potential of the working electrode was meticulously monitored using V7-27 DC voltmeter. Specifically, a platinum cylindrical cathode, boasting a surface area of 20 $cm^2$, was designated as the working electrode, while a platinum rod served as the anode. The working electrode's potential was referenced using $Ag/AgNO_3$ electrode in $CH_3CN$. The catholyte, comprising a saturated solution of $PyHBF_4$ in $CH_3CN$, played a pivotal role in the electrochemical process. Throughout the electrolysis procedure, a magnetic stirrer ensured thorough mixing of the electrolyte. To maintain optimal conditions, the entire process unfolded within a continuous flow of inert gas, expertly designed to eliminate any traces of oxygen or other gaseous impurities.

### 3.7. General Electrolysis Procedure

The electrochemical cell, completed with a magnetic stir bar, was carefully loaded with 1.38 mmol of quinoline *N*-oxide, 0.14 mmol of copper acetate, 1.65 (or 4.14) mmol of morpholine, and 5 mmol of $K_3PO_4$, all dissolved in 20 mL of acetonitrile, within an argon-filled environment at 25 °C. The resulting mixture underwent stirring at room temperature. The electrolysis duration was typically 75 min for 2F electricity, 115 min for 3F, or 150 min for 4F electricity. Upon completion of the electrolysis process, any precipitated $K_3PO_4$ was meticulously filtered out. The reaction mixture was subjected to evaporation using a rotary evaporator, followed by a chloroform wash. Subsequently, the residue obtained after chloroform removal was purified chromatographically using a column packed with silica and hexane-ethyl acetate eluent.

**4-(Quinolin-4-yl)morpholine (3).** Yellow solid, m.p. 83–85 °C. Yield 0.19 g (64%). $^1$H NMR (399.9 MHz, CDCl$_3$): δ 8.71 (d, *J* = 8.8 Hz, 1H); 8.06 (d, *J* = 7.4 Hz, 1H); 7.85 (d, *J* = 8.1 Hz, 1H); 7.62 (t, *J* = 6.9 Hz, 1H); 7.45 (t, *J* = 7.1 Hz, 1H); 7.29 (d, *J* = 6.4 Hz, 1H); 4.04 (t, *J* = 4.7 Hz, 4H); 3.23 (t, *J* = 4.8 Hz, 4H). $^{13}$C NMR (100.6 MHz, CDCl$_3$), δ ppm: 158.14 ($C_i$), 149.36 (CH), 141.95 ($C_i$), 130.95 (CH), 129.25 (CH), 128.63 (CH), 127.29 ($C_i$), 126.66 (CH), 118.09 (CH), 66.30 ($CH_2$), 53.85 ($CH_2$). MS (ESI), *m/z*: 215.1 [M + 1]$^+$. Anal. calc. (%): C, 72.87; H, 6.59; N, 13.07. $C_{13}H_{14}N_2O$. Found (%): C, 72.69; H, 6.35; N, 13.12.

**4,4′-(Quinoline-2,4-diyl)dimorpholine (4).** Yellow solid, m.p. 165–167 °C. Yield 0.21 g (52%). $^1$H NMR (399.9 MHz, CDCl$_3$): δ 8.28 (d, *J* = 7.5 Hz, 1H); 8.08 (m, 2H); 7.79 (d, *J* = 6.9 Hz, 1H); 7.55 (t, *J* = 7.1 Hz, 1H); 3.71 (t, *J* = 4.5 Hz, 4H); 3.67 (t, *J* = 4.8 Hz, 4H); 3.59 (t, *J* = 4.8 Hz, 4H); 3.40 (t, *J* = 4.85 Hz, 4H). $^{13}$C NMR (100.6 MHz, CDCl$_3$), δ ppm: 159.22 ($C_i$), 152.74 ($C_i$), 141.32 ($C_i$), 129.83 (CH), 128.86 (CH), 128.77 (CH), 123.98 ($C_i$), 123.37 (CH), 109.47 (CH), 61.32. ($CH_2$), 61.07 ($CH_2$), 52.80 ($CH_2$), 52.74 ($CH_2$). MS (ESI), *m/z*: 300.2 [M + 1]$^+$. Anal. calc. (%): C, 68.20; H, 7.07; N, 14.04. $C_{17}H_{21}N_3O_2$. Found (%): C, 68.01; H, 6.85; N, 14.14.

**2-Morpholinoquinoline 1-oxide (5).** Yellow solid, m.p. 126–128 °C. Yield 0.25 g (80%). $^1$H NMR (399.9 MHz, CDCl$_3$): δ 8.82 (d, *J* = 8.6 Hz, 1H); 8.39 (d, *J* = 6.1 Hz, 1H); 7.95 (d, *J* = 8.1 Hz, 1H); 7.86–7.82 (m, 2H); 7.55 (t, *J* = 8.1 Hz, 1H); 3.92 (t, *J* = 4.7 Hz, 4H); 3.56 (t, *J* = 4.7 Hz, 4H). $^{13}$C NMR (100.6 MHz, CDCl$_3$), δ ppm: 152.19 ($C_i$), 141.92 ($C_i$), 130.91 (CH), 129.21 (CH), 128.59 (CH), 126.63 (CH), 124.13 ($C_i$), 120.12 (CH), 118.51 (CH), 68.44 ($CH_2$), 46.85 ($CH_2$). MS (ESI), *m/z*: 231.1 [M + 1]$^+$. Anal. calc. (%): C, 67.81; H, 6.13; N, 12.17; $C_{13}H_{14}N_2O_2$; Found (%): C, 67.92; H, 6.25; N, 12.12.

**2-Morpholinoquinoline (6).** Pale yellow solid, m.p. 87–89 °C. Yield 0.19 g (67%). $^1$H NMR (399.9 MHz, CDCl$_3$): δ 8.18 (d, *J* = 8.68 Hz, 1H); 7.99 (d, *J* = 6.0 Hz, 1H); 7.75 (d, *J* = 6.0 Hz, 1H); 7.53 (t, *J* = 7.9 Hz, 1H); 7.40 (d, *J* = 8.0 Hz, 1H); 6.92 (d, *J* = 7.6 Hz, 1H); 3.86 (t, *J* = 4.6 Hz, 4H); 3.68 (t, *J* = 4.6 Hz, 4H). $^{13}$C NMR (100.6 MHz, CDCl$_3$), δ ppm: 154.14 ($C_i$), 141.90 ($C_i$), 136.06 (CH), 129.19 (CH), 128.57 (CH), 124.11 ($C_i$), 121.41 (CH), 120.10 (CH),

110.24 (CH), 66.43 ($CH_2$), 45.56 ($CH_2$). MS (ESI), *m/z*: 215.3 [M + 1]$^+$. Anal. calc. (%): C, 72.87; H, 6.59; N, 13.07; $C_{13}H_{14}N_2O$; Found (%): C, 72.76; H, 6.48; N, 13.10.

**Morpholin-4-ol (7).** $^1$H NMR (399.9 MHz, CDCl$_3$): δ 8.08 (s, 1H); 3.70 (t, *J* = 4.7 Hz, 2H); 3.66 (t, *J* = 4.6 Hz, 2H); 3.58 (t, *J* = 4.6 Hz, 4H); 3.68 (t, *J* = 4.7 Hz, 4H). MS (ESI), *m/z*: 104.1 [M + 1]$^+$. Anal. calc. (%): C, 46.59; H, 8.80; N, 13.58. $C_4H_9NO_2$; Found (%): C, 46.77; H, 8.49; N, 13.45.

**4-(Isoquinolin-1-yl)morpholine (8).** $^1$H NMR (399.9 MHz, CDCl$_3$): δ 8.22 (d, *J* = 8.4Hz, 1H); 8.14 (t, *J* = 6.1 Hz, 1H); 7.94–7.88 (m, 2H); 7.78 (t, *J* = 7.3 Hz, 1H); 7.49 (d, *J* = 6.6 Hz, 1H); 4.03 (m, 4H); 3.88 (m, 4H). $^{13}$C NMR (100.6 MHz, CDCl$_3$), δ ppm: 159.69 ($C_i$), 139.69 ($C_i$), 137.38 (CH), 130.13 (CH), 127.30 (CH), 126.28 (CH), 125.09 ($C_i$), 122.74 (CH), 117.05 (CH), 68.01 ($CH_2$), 52.35 ($CH_2$). MS (ESI), *m/z*: 215.1 [M + 1]$^+$. Anal. calc. (%): C, 72.87; H, 6.59; N, 13.07; $C_{13}H_{14}N_2O$; Found (%): C, 72.69; H, 6.42; N, 13.15.

## 4. Conclusions

In conclusion, an efficient method for the preparation of quinoline amino-derivatives from quinoline *N*-oxides and morpholine by Cu(II)-catalyzed selective C–H bond functionalization was established. This catalytic transformation could be controlled by solvent, and $CH_3CN$ favors the formation of an ortho-substituted product in the first stage with a 1:1 ratio of reagents, but a para-substituted product is formed in methylene chloride. For the first time, the morpholine radical was detected by EPR at room temperature in the absence of any external oxidizers, irradiation, and trapping agents. This electrochemical method offers a facile route to aminoquinolines (or oxides) at room temperature, in the absence of specially added external oxidizing agents, suggests the use of morpholine itself and not its derivatives (such as O-benzoylhydroxylamine). This is an atom-economical, low-waste synthesis option with good product yields, the nature of which can be controlled by changing the ratio of reagents, solvent, and the amount of electricity passed.

**Supplementary Materials:** The following supporting information can be downloaded at https://www.mdpi.com/article/10.3390/catal13091279/s1, Table S1: Electrochemical cross-coupling products of morpholine and *N*-oxide, 60 mA, Figure S1: Spectrum 1H NMR of 4-(Quinolin-4-yl)morpholine (3) in CDCl$_3$, Figure S2: Spectrum $^{13}$C NMR of 4-(Quinolin-4-yl)morpholine (3) in CDCl$_3$, Figure S3: Spectrum $^1$H NMR of 4,4′-(Quinoline-2,4-diyl)dimorpholine (4) in CDCl$_3$, Figure S4: Spectrum $^{13}$C NMR of 4,4′-(Quinoline-2,4-diyl)dimorpholine (4) in CDCl$_3$, Figure S5: Spectrum $^1$H NMR of 2-morpholinoquinoline 1-oxide (5) in CDCl$_3$, Figure S6: Spectrum $^{13}$C NMR of 2-morpholinoquinoline 1-oxide (5) in CDCl$_3$, Figure S7: Spectrum $^1$H NMR of 2-morpholinoquinoline (6) in CDCl$_3$, Figure S8: Spectrum $^{13}$C NMR of 2-morpholinoquinoline (6) in CDCl$_3$; Figure S9: Spectrum $^1$H NMR of morpholine-1-oxide (7) in CDCl$_3$, Figure S10: Spectrum $^1$H NMR 4-(isoquinolin-1-yl)morpholine (8) in CDCl$_3$, Figure S11: Spectrum $^{13}$C NMR of 4-(isoquinolin-1-yl)morpholine (8) in CDCl$_3$. References [28–35] are cited in the Supplementary Materials.

**Author Contributions:** Conceptualization, Y.H.B.; methodology, T.V.G.; investigation, E.L.D.; EPR data curation, K.V.K.; writing—original draft preparation, Y.H.B.; writing—review and editing, T.V.G., E.L.G., E.L.D. and O.G.S.; supervision and project administration, Y.H.B. All authors have read and agreed to the published version of the manuscript.

**Funding:** This research was funded by the Russian Science Foundation grant no. 22-13-00017.

**Data Availability Statement:** All data are available in the manuscript or upon request to the corresponding author.

**Acknowledgments:** The authors thank the Assigned Spectral-Analytical Center of the FRC Kazan Scientific Center of the RAS for the provided research equipment (electrochemical, spectral, etc.).

**Conflicts of Interest:** The authors declare no conflict of interest.

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
