# Peer review of "Morpholine Radical in the Electrochemical Reaction with Quinoline N-Oxide"

_catalysts, doi:10.3390/catal13091279_

Round 1
Reviewer 1 Report
Reviewer’s comments
The current manuscript describes developed electrochemical method by using Cu(OAc)2 as catalyst, for synthesis of 4-aminoquinolines N-oxides in CH2Cl2 or 2-aminoquinolines N-oxides in CH3CN in good yields. 1HNMR, 13CNMR and EPR spectroscopy techniques as well as a cyclic voltammetry were used to characterize the separated products. The advantages of the applied method are simple, easy to perform at mild conditions, room temperature, use of morpholine instead of its derivatives, and ability to control the process when the electrolysis conditions change.
From my point of view, this work is good, the topic is interesting, and it was implemented professionally, reflecting the team's knowledge of the scientific methodology of the current study. Accordingly, I recommend accepting this work in its current form for publication in the Journal Catalysts.
Author Response
Thank you for your review!
Reviewer 2 Report
The manuscript by Budnikova and co-workers describe a Cu(II)-mediated coupling of morpholine with quinoline-N-oxide under electrochemical conditions. The regioselectivity of the reaction appears to depend strongly on the solvent employed: in dichloromethane, a 4-substituted product is produced, whereas acetonitrile favours 2-substitution. In excess morpholine, a disubstituted compound is formed. The reaction mechanism was probed by CV and ESR studies. It was proposed that in dichloromethane, the reaction proceeds via morpholine radical, whereas such radical species were not observed in acetonitrile, suggesting a different reaction manifold. In both solvents, a C-H insertion of Cu into quinoline N-oxide was suggested, however, this does not look convincing. The author should provide literature examples supporting such an assumption. A radical mechanism seems more likely. On page 5, lines 123-124, the authors claimed that dichloromethane can reduce the N-O bond in quinoline N-oxide, which is highly unlikely. In fact, the electrochemical reduction of N-oxides is known: a simple literature search identified at least two research papers, Synlett 2019, 30, 1219–1221 and EurJOC, 2020 (22), 3117-3119, which should be cited. My other concern is that the manuscript presented a single example of the reaction, the reaction scope has not been investigated. For publication, a few other aromatic N-oxides should be tested.
Overall, the report is interesting but there are some issues listed above, including the lack of scope, which prevent me at this point from recommending this manuscript for publication.
some modest editing is needed, primarily for sentence structure
Author Response
The manuscript by Budnikova and co-workers describe a Cu(II)-mediated coupling of morpholine with quinoline-N-oxide under electrochemical conditions. The regioselectivity of the reaction appears to depend strongly on the solvent employed: in dichloromethane, a 4-substituted product is produced, whereas acetonitrile favours 2-substitution. In excess morpholine, a disubstituted compound is formed. The reaction mechanism was probed by CV and ESR studies. It was proposed that in dichloromethane, the reaction proceeds via morpholine radical, whereas such radical species were not observed in acetonitrile, suggesting a different reaction manifold.
In both solvents, a C-H insertion of Cu into quinoline N-oxide was suggested, however, this does not look convincing. The author should provide literature examples supporting such an assumption.
Reply:
Below we present literary examples supporting such an assumption. Indeed, in many papers on the activation of the C-H bond with the participation of copper catalysts, the formation of a copper-carbon bond, or metallization of quinoline N-oxide (or another heterocycle) with Cu(OAc)2, is assumed. However, no one succeeded in isolating and characterizing it in its pure form, as well as we, although we tried to obtain such sigma organometallic compounds with copper in different oxidation states. Therefore, like all other authors below the list, we limited ourselves to the assumption of their formation, which explains the selectivity of the reaction, the formation of a substitution product in position 2 of quinoline N-oxide:
- Copper-Catalyzed Intermolecular Dehydrogenative Amidation/Amination of Quinoline N‑Oxides with Lactams/Cyclamines, Gang Li, Chunqi Jia, Kai Sun. ORGANIC LETTERS, 2013, Vol. 15, No. 20, 5198–5201. 10.1021/ol402324v
- Sulfonylation of Quinoline N-Oxides with Aryl Sulfonyl Chlorides via Copper-Catalyzed C–H Bonds Activation. Wu, Z. Y.; Song, H. Y.; Cui, X. L.; Pi, C.; Du, W. W.; Wu, Y. J. Org. Lett. 2013, 15, 1270- 1270–1273. https://doi.org/10.1021/ol400178k
- Copper(II)-catalyzed electrophilic amination of quinoline N-oxides with O-benzoyl hydroxylamines. Org. Biomol. Chem., 2015, 13, 3207 –3210 DOI: 10.1039/c5ob00135h
«The metallization of quinoline N-oxide with species A then generated arylcopper B, which was added to O-benzoyl hydroxylamine to form the crucial intermediate C.»……..
- Copper-Catalyzed Direct Amination of Quinoline N‑Oxides via C−H Bond Activation under Mild Conditions, Chongwei Zhu, Meiling Yi, Donghui Wei, Xuan Chen, Yangjie Wu, Xiuling Cui. Org. Lett. 2014, 16, 1840−1843. doi.org/10.1021/ol500183w |
«First, the 2-carbon of the quinoline N-oxide was attacked by copper and afforded intermediate I (Cu-arene)»….
- Zhang, M. L.; Zhang, S. H.; Liu, M. C.; Cheng, J. Chem. Commun. 2011, 47, 11522. (C-arylation of a benzo[d]oxazole)
- Wu, X. F.; Anbarasan, P.; Neumann, H.; Beller, M. Angew. Chem., Int. Ed. 2010, 49, 7316.
A radical mechanism seems more likely.
Reply:
The radical mechanism is most likely in dichloromethane, where we have proven the formation of the morpholine radical. In other cases, there is no reason to assert a radical mechanism, which, as a rule, is not very selective.
On page 5, lines 123-124, the authors claimed that dichloromethane can reduce the N-O bond in quinoline N-oxide, which is highly unlikely.
Reply:
This is not ours, but the literary statement of the authors from the reference [13, Wang, Z.; Han, M.-Y.; Li, P.; Wang, L. Copper-Catalyzed Deoxygenative C-2 Amination of Quinoline N-Oxides. European J Org Chem 2018, 2018, 5954–5960, doi:10.1002/ejoc.201800963], which describes the use of dichloromethane as a reducing agent: "Abstract: An unprecedented reaction between quinoline N-oxides with O-benzoylhydroxylamines was developed by using CuCl as catalyst, generating deoxygenative products of 2-aminoquinolines in good yields. 1,2-Dichloroethane (DCE) assolvent was also served as a reducing agent to cleave the N–O bond with no additional reductant needed in the reaction.” Although, we didn’t study this particular stage ourselves, this seems as a likely scenario.
In fact, the electrochemical reduction of N-oxides is known: a simple literature search identified at least two research papers,Synlett 2019, 30, 1219–1221 and EurJOC, 2020 (22), 3117-3119, which should be cited. My other concern is that the manuscript presented a single example of the reaction, the reaction scope has not been investigated. For publication, a few other aromatic N-oxides should be tested.
Reply:
Examples of electrochemical deoxygenation of N-heteroaromatic N-oxides are known, but these are cathodic electrochemical reduction reactions. In our case, we are studying anodic processes, so the suggested references, in our opinion, are not very suitable for our reactions and do not explain anything. However, taking into account the comments of the reviewer, we have added them to the list of cited literature:
- Electrochemical Deoxygenation of N-Heteroaromatic N-Oxides. P. Xu , H.-C. Xu. Synlett 2019, 30, 1219–1221 DOI: 10.1055/s-0037-1611541
- Synergy of anodic oxidation and cathodic reduction leads to electrochemical deoxygenative C2 arylation of quinoline N-oxides. Yong Yuan, Minbao Jiang, Tao Wang, Yunkui Xiong, Jun Li, Huijiao Guoac and Aiwen Lei. Commun., 2019, 55, 11091 DOI: 10.1039/c9cc05841a
The reference EurJOC, 2020 (22), 3117-3119 is incorrect, this article does not exist.
We tested other aromatic N-oxides and the results are listed in Supp Information. Of the studied N-oxides, iso-quinoline N-oxide showed high yields in the morpholination reaction, while others were either non-reactive or the reaction proceeded non-selectively (SI, Table S1).
Overall, the report is interesting but there are some issues listed above, including the lack of scope, which prevent me at this point from recommending this manuscript for publication.
Comments on the Quality of English Language: some modest editing is needed, primarily for sentence structure
Reply: Thank you, we have edited the text.

Reviewer 3 Report
This manuscript conveys the work on electrochemical C-H amination of quinoline N-oxide to aminated quinoline products. The authors have made efforts in investigating the catalytic mechanisms by EPR, cyclic voltage experiments. The results are beneficial for understanding the free radical C-H amination reactions for the synthesis of functionalized quinolines. The work is recommended for publication after revision by addressing the following issues.
1. As a method for synthesizing quinoline derivative, the value of such method should be properly discussed based on recent advances: J. Org. Chem. 2022, 87, 16343-16350; Chin. J. Org. Chem. 2022, 42, 3721-3729; Org. Lett. 2019, 21, 3600-3605 etc.
2. In the proposed mechanisms, the reaction in electrodes should be shown clearly.
3. To make the mechanism clear, please provide the valent of the Cu center in each step.
4. While the results show that the solvent make different in providing different products (DCM vs MeCN), than the reasons for such solvent-based selectivity should be carefully discussed.
5. The authors are suggested to expand the electrocatalytic amination to the reactions of other cyclic and acyclic secondary amines.
No
Author Response
This manuscript conveys the work on electrochemical C-H amination of quinoline N-oxide to aminated quinoline products. The authors have made efforts in investigating the catalytic mechanisms by EPR, cyclic voltage experiments. The results are beneficial for understanding the free radical C-H amination reactions for the synthesis of functionalized quinolines. The work is recommended for publication after revision by addressing the following issues.
- As a method for synthesizing quinoline derivative, the value of such method should be properly discussed based on recent advances: Org. Chem. 2022, 87, 16343-16350; Chin. J. Org. Chem. 2022, 42, 3721-3729; Org. Lett. 2019, 21, 3600-3605 etc.
Reply: Done. We added these references.
- In the proposed mechanisms, the reaction in electrodes should be shown clearly.
Reply: Done
- To make the mechanism clear, please provide the valent of the Cu center in each step.
Reply: Done
- While the results show that the solvent make different in providing different products (DCM vs MeCN), than the reasons for such solvent-based selectivity should be carefully discussed.
Reply: Reaction mechanism, as it seems, depends on solvent’s nature, leading to different products. ESR studies show the formation of morpholine radical in CH2Cl2. We added discussions of such behavior in description of mechanisms, as well as discussions of participation of DCM in deoxygenation of quinoline N-oxide, based on examples in literature.
- The authors are suggested to expand the electrocatalytic amination to the reactions of other cyclic and acyclic secondary amines.
Reply: Thanks for the suggestion, however, each other cyclic and acyclic secondary amines has a different reactivity, different redox properties, and requires separate detailed studies, which we will continue in the future. We have devoted this article to morpholine and its properties in redox reactions, as not very well studied. In addition, there is a huge amount of other work on the reactions of other cyclic and acyclic secondary amines, this is a well-plowed field.
Round 2
Reviewer 2 Report
The revised manuscript addressed most of the issues raised by the reviewers. Just a few more comments. As a general rule, I would strongly discourage citing unproven mechanisms/processes as evidence in support of the statements in the discussion, at least without a clear disclaimer (which authors indeed inserted for the C-H metalation step). This will avoid the proliferation of speculative/erroneous mechanistic statements. In this context, in Scheme 3 the authors still use dichloromethane as a reducing agent for the N-oxide, which is wrong – this solvent, and other chlorinated solvents, are commonly used as the reaction medium for aromatic N-oxides with no observed deoxygenation, the reduction is more likely to occur on the cathode, even in the divided cells under stirring. For the suggested report on the electrochemical reduction of N-oxides, the wrong pages were given mistakenly, the correct reference is as follows: Eur J Org Chem, 2020, 3317.
Other than that, I am happy to recommend publication.
some minor grammar inconsistencies
Author Response
There is no reaction in the world with absolutely, 100% proven mechanisms at all stages. Nevertheless, discussion of possible paths and probable intermediates is always welcome, and the search for stabilization of these intermediates continues, although not always crowned with success. Since it is known that the copper–carbon bond generally exists and can be formed in various reactions where such intermediates have been isolated and confirmed, including by X-ray diffraction analysis [for example, J. Am. Chem. Soc. 2017, 139, 27, 9112-9115], it is permissible to assume related intermediates in other catalytic cycles. In the absence of copper, the reaction does not proceed at all, that is, the simple generation of a morpholine radical at the anode does not lead to success. Taking into account the comments of the reviewer, we have put in square brackets the putative intermediates with copper-carbon bonds.
In Scheme 3, we removed dichloromethane from the last step. However, we cite an article [13 : «1,2-Dichloroethane (DCE) as solvent also served as reducing agent to cleave the N–O bond with no additional reductant needed in the reaction» ] where the possibility of its participation in the reaction of deoxygenation of N-oxides in the presence of copper salts is proved. But since deoxygenation is also observed in acetonitrile with Cu-catalyst, although to a lesser extent, we will not focus on the degree of influence of the solvent on this stage, although it exists, and the products generally depend on the nature of the solvent. Regarding the reviewer's assumption about cathodic deoxygenation: there is no reason to assert the latter in the processes described. The reaction does not take place in a undivided cell. Diffusion of N-oxide into the cathode space through the membrane was not recorded in our conditions. In addition, in the cathode department we use a background salt that is reduced much faster (-1.6 V) than the N-oxide (-2.3 V vs. Fc/Fc+, see figure below). That is, the reduction potential of the N-oxide is not reached in the presence of this easily reduced electrolyte, which is also present in excesses. We have added the reference Eur J Org Chem, 2020, 3317 to the list of references, although the conditions in it are fundamentally different.
